# Stepwise Imperatives for Improving the Protection of Animals in Research and Education in Canada

**DOI:** 10.3390/ani14192755

**Published:** 2024-09-24

**Authors:** Kyrstin Lavelle, Karim Fouad, Judy Illes

**Affiliations:** 1Neuroethics Canada, Division of Neurology, Department of Medicine, University of British Columbia, Vancouver, BC V6T 2B5, Canada; klavelle@student.ubc.ca; 2Department of Physical Therapy, Neuroscience and Mental Health Institute, Faculty of Rehabilitation Medicine, University of Alberta, Edmonton, AB T6G 2G4, Canada; kfouad@ualberta.ca

**Keywords:** animal, research, ethics

## Abstract

**Simple Summary:**

Canada uses a decentralized approach to regulate animal use in science that can weaken protections, create confusion for compliance, and complicate enforcement. We propose a stepwise approach to centralizing animal welfare laws, beginning with an enhancement of the existing 3Rs (Replacement, Refinement, and Reduction) framework by adding two new Rs: Reflection and Responsiveness. The two new Rs encourage the continuous evaluation of research progress, timely action based on new findings, and the publication of all data. We suggest that proactive consideration of all five Rs in research and education would immediately improve the welfare of animals used in science across Canada while restructuring towards a more centralized approach is explored.

**Abstract:**

In this paper, we review the standard-setting tools of different levels of government in Canada for overseeing the care of animals used in science against a landscape of other international efforts. We find regulatory inconsistencies, argue that the related shortcomings are detrimental to the level of care afforded to animals, and offer suggestions for a centralized and proactive approach that could close the existing gaps. Given the resources, cost, and time it would take to transform the current system into a single cohesive one, the proposed approach is a stepwise one, and begins with the addition of two new Rs—Reflection and Responsiveness—to the existing 3Rs framework: Replacement, Refinement, and Reduction. Reflection emphasizes more continuous and specific attention to progress in the research pathway as it applies to animals than is currently required by institutional review and reporting; Responsiveness speaks to the immediate action that researchers can take responsively to that ongoing evaluation.

## 1. Introduction

Animal models are a significant component of scientific discovery, utilized by both ancient philosophers and modern researchers alike, and have been an essential component in the exploration and understanding of complex biological processes, diseases, and treatments [1,2]. Aristotle, among others, believed that animals were intended for human use. He concluded that “as nature makes nothing purposeless or in vain, all animals must have been made by nature for the sake of man” [3] (Politics, 1, VIII, 11–12). This view was widely accepted and remained influential for a long time despite the emergence of opposing views around the late nineteenth century. The turning point centered around the controversy of vivisection and whether scientific progress justified the suffering of animals [3]. Organized anti-vivisection movements gained momentum, leading to the passage of the first animal protection laws, such as the UK’s Cruelty to Animals Act of 1876. This marked a shift towards recognizing animal welfare concerns in scientific research, though it did not halt animal experimentation [4]. At the beginning of the twentieth century, the rise of industrial biomedicine made animal models crucial checkpoints for research such as pharmaceutical development. At the same time, terms such as “speciesism” were introduced, suggesting that the consideration of one species as better than or of more value [5].

There has been a tenfold increase in publications about animal sentience from 1990 to 2011 [6], showing a surge in attention towards the ethical use of animals in biomedical and behavioral research. This shift in awareness has been accompanied by significant changes in practice. For example, in the USA, research with chimpanzees has been discontinued and similar measures have been implemented for great apes in the European Union (EU) [7]. Similarly, in 2013, the EU banned animal testing for cosmetics to protect animals from unnecessary pain and injuries, an initiative that India and Australia followed [8]. The Government of Canada also banned cosmetic testing on animals, citing increased concern for the well-being of animals in 2023 [9]. Yet, conflicting regulations continue to present major challenges. Knight et al. (2021), for example, found that even with the Cosmetic Regulation ban in the EU, some ingredients still undergo in vivo testing to fulfill the requirements of another regulation called REACH (registration, evaluation, authorization, and restriction of chemicals) [10].

The 3Rs—Replacement, Refinement, and Reduction—have served as an ethical framework for animal research since 1959, when Russell and Burch published *The Principles of Humane Experimental Technique* [11]. Replacement refers to methods that avoid or replace the use of sentient animals in a study in which they would otherwise have been used; Refinement refers to modifications to husbandry or experimental procedures that minimize pain and distress for an animal; Reduction refers to any strategy that results in fewer animals being used to obtain sufficient data to answer a research question while maximizing the information obtained per animal. Scientists are required to comply with the laws and regulations of their jurisdictions, many of which incorporate the 3Rs into their guidance schemes. Many countries, including Canada, include a combination of legislative tools, such as laws, and non-legislative tools such as the guidelines and policies published by the Canadian Council on Animal Care (CCAC) [12]. Unlike the EU which has standardized protection for animals in research through Directive 2010/63/EU, Canada has one of the most decentralized approaches in the world to protect animals in science. Laws are spread between national and subnational (provincial and territorial) levels. This decentralization is the source of inconsistencies in the application of standards [13], with direct implications for ethical practice in Canadian research and educational settings that impacts the numbers of animals used and reported, choices of animal models, and continuity of care for animals.

To improve the protection of animals across all jurisdictions, we advocate for an updated legislative model for Canada. We recognize that transforming animal care in research in this way (i.e., centralizing it) would involve the coordination of both levels of government and a constitutional change [13]. As this is a high bar that will be difficult to reach in short course, we propose a stepwise approach to this goal with the immediate consideration of two new Rs, Reflection and Responsiveness, in addition to the existing three.

## 2. Overview of Animal Research Regulations in Canada

Regulations and guidance for animal research are present at both the federal and provincial/territorial levels in Canada. General welfare protection for all animals are seen in the Canadian Criminal Code, the Health of Animals Act, and in the mandates of the Canadian Food Inspection Agency (CFIA). However, the Criminal Code does not mention research, neither as an activity that has an exemption to permit incidental suffering or injury to animals, nor as an activity that is regulated elsewhere. Federal legislative power is limited, because animals are considered property, and property historically falls within the jurisdiction of the provincial and territorial governments [13,14]. At this level, each province and the Yukon maintain legislation pertaining to animal welfare. Their regulations cover similar themes and share the objective of protecting all animals from abuse. Common topics include duties of care, penalties, enforcement, transportation, referenced standards, and exemptions. Seven provinces—Alberta, Manitoba, New Brunswick, Newfoundland and Labrador, Nova Scotia, Ontario, and Prince Edward Island—have specific legislation related to animals in research and teaching.

The CCAC [15] is a national body that provides oversight for animal-based scientific activities for institutions that participate in CCAC certification. Its mission is to ensure that animal-based science takes place only when necessary and that the animals in experimental studies receive optimal care according to high-quality, research-informed standards. CCAC guidelines consider animal use in seven different broad categories: fundamental science, environmental science, human medicine and well-being, veterinary medicine and animal welfare, food production, regulatory testing, and education. Established in 1968, the CCAC receives 50% of its funding from the Canadian Institute of Health Research (CIHR) and the Natural Sciences and Engineering Research Council (NSERC). The other half is generated through program participation fees charged to certified institutions, including a small portion through revenue from conferences and other programs [16]. The CCAC ensures compliance through announced inspections. According to the CCAC, 3,521,143 animals were used in research, teaching, and testing in Canada in 2022 [17].

## 3. Tensions

While the current Canadian system may seem comprehensive, small compounding differences between provinces and territories create a landscape of confusing definitions and exceptions. For instance, Manitoba’s Animal Care Act defines an animal, in Section 2, as “a non-human living being with a developed nervous system” [18], whereas British Columbia defines an animal, in Section 1 of its Health of Animals Act, as “a species of the animal kingdom, and any organism prescribed as an animal” [19]. The semantics are similar, but the inconsistency exists at the most basic level: according to Manitoba’s definition, the fruit fly (Drosophila melanogaster), used extensively in genetic research, may not qualify as an animal since it does not have a developed nervous system; under British Columbia’s broader definition that encompasses any species of the animal kingdom and prescribed organisms, the fruit fly could be included.

Inconsistencies are also seen within provinces and territories, as the composition of each research institute’s Animal Care Committee (ACC) varies. Vardigans et al. (2019) reviewed animal use protocol forms required by institutional ACCs, and found that the information provided often failed to meet the CCAC’s expected standards [20]. They reported that the forms failed to provide enough information for the ACCs to make informed decisions that are consistent with the 3Rs. Common issues include ambiguous wording, a lack of specific guidance, and failure to ask for detailed information in the search for alternatives. These issues compound, diluting the efficacy and efficiency of the review process at each institution. Penalties on investigators might even ensue.

Table 1 details which provinces and territories explicitly mention animals in research or teaching within their legislation. It highlights which provinces have specific laws for animals, references specific CCAC guidelines, and emphasizes Ontario’s standalone act. In addition to inconsistent and redundant legislation, three provinces and three territories lack direct legislation on animal research altogether, and rely solely on the CCAC guidelines as suggestions and not laws. This gap allows an act to be legal in one province while being illegal in another. For instance, Ontario’s stricter regulations may lead to more penalties compared to Saskatchewan, which lacks direct legislation for research animals. Penalties for violations are also different between provinces.

Inconsistent monitoring and the incorporation of guidelines from the CCAC across provinces further exacerbate challenges. Table 2 highlights these inconsistencies by showing the five provinces that explicitly reference specific CCAC standards for researchers in their animal welfare legislation. For example, Alberta’s Animal Protection Regulation Section 2.1 states, “a person who owns or has custody, care or control of an animal for research activities must comply with the following Canadian Council on Animal Care documents” [21] referring to a comprehensive list of CCAC standards. By contrast, Newfoundland and Labrador’s regulations also incorporate some CCAC guidelines into law, but they mandate fewer standards, and not the same ones as Alberta. Table 2 only cites the titles, and it is important to note that provinces may cite different volume numbers or publication years when referring to the same larger document. Since the CCAC updates its guidelines more frequently than legislation is passed, provinces might inadvertently reference outdated versions.

The CCAC has the authority to revoke certification for non-compliance, which would be reported to CIHR and NSERC who could suspend or revoke funding [22]. Institutions that have significant program gaps may be placed on probation, and must meet the CCAC requirements to regain full certification. It is important to note that the current system is opt-in. Privately funded institutions or organizations not receiving CIHR or NSERC funding, or those receiving grant or contract funding from other federal or provincial departments are not always obliged to be certified by CCAC unless specified in the contract. Such organizations may choose to seek CCAC certification for public trust, but as there is no requirement to do so, they effectively self-regulate their animal research. The exact number of animals used in private laboratories is difficult to estimate due to the lack of mandatory reporting requirements and the absence of a legislative pathway to understand the activities within these privately funded research facilities.

**Table 2 animals-14-02755-t002:** Relevant CCAC guidelines referenced in animal welfare regulations by provinces and territories that have them. [Volume number and publication year may vary. Guidelines listed by all 5 provinces are not included. Regulations not directly related to research and teaching were excluded.].

	Animal Use Protocol Review	Choosing an Appropriate endpoint in Experiments Using Animals for Research, Teaching and Testing	Institutional Animal User Training Program	Antibody Production	Care and Use of Wildlife	Laboratory Animal Facilities-Characteristics, Design and Development	Care and Use of Fish in Research, Teaching and Testing	Terms of Reference for Animal Care Committees	Policy Statement on: Ethics of Animal Investigation	Social and Behavioural Requirements of Experimental Animals (SEBREA)	Acceptable Immunological Procedures	Categories Of Invasiveness In Animal Experiments	Definitions of recommendations Made in CCAC Reports	Compliance and Non-Compliance	Importance of Independent Peer Review of the Scientific merit of Animal-Based Research Projects	Animal-Based Projects Involving Two or More Institutions	Transgenic Animals	Care and Use of Farm Animals in Research, Teaching and Testing, Published by the CCAC	Euthanasia of Animals Used in Science	Confidentiality of Assessment Information	Certification of Animal Care and Use Programs	Procurement of Animals Used in Science
Alberta [21]																						
Manitoba [23]																						
New Brunswick [24]																						
Newfoundland and Labrador [25]																						
Prince Edward Island [26]																						

## 4. A Stepwise Call to Action

Echoing past calls from Fraser et al. and Black et al. [14,22], we ultimately advocate for the creation of an Animals in Science Act. Such legislation would centralize benchmarks, and implementation and outcome metrics for animal use in science and education across Canada. To address these issues, the Act would standardize definitions, elevate all the CCAC guidelines to law, include provisions for monitoring and addressing violations in both the private and public sectors, establish consistent standards, and streamline enforcement. We believe that such an Act would reduce animal use in research and training, enhance care, improve scientific quality, minimize costs and redundancy, and increase the regulation of the private sector.

We acknowledge the challenges to creating such an Act. First, elevating guidelines to laws can be complicated. The current guidelines offered by the CCAC contain “should” and “must” statements that indicate best practices or mandatory standards. They outline best practices and encourage improvement (Refinement) in incremental steps. However, laws and regulations are cut and dry. They become a universal standard. Compliance is a requirement. An Act should standardize the highest level of care without discouraging best practices and improvement. Second, current assessments would become inspections. The collaborative, collegial approach currently provided by assessment panels, which include peers and community representatives, would have to be replaced by an official system that identifies deficiencies and ways to correct them. It is crucial that this type of inspection does not hinder information exchange and practical solutions to common problems that currently occur during assessments.

Immediate, simpler steps can be taken to enhance animal protection in science within Canada’s fractured legislative landscape. We recommend an upgrade to the three guiding Rs who, like others, have called for innovative methods to improve them [27,28,29]. We recognize the positive impact of initiatives such as the ARRIVE guidelines, first published in 2010, which set important standards for the minimum information required in animal study publications [30]. However, the focus of the ARRIVE guidelines and the current 3R framework is on the design phase of research. The proposed upgrade expands the traditional 3Rs framework, and introduces two additional Rs—Reflection and Responsiveness—for application across experimental and educational continuums. [31]. Reflection in this model complements the other Rs by encouraging active engagement around progress in research. It focuses on (1) successes that may lead to the need for fewer animal data without compromising statistical power, and (2) the reporting of (and education about) negative results and evidence of futility may signal the need for early study termination or prevent against duplications.

The fifth R—Responsiveness—refers to the reflexive responsibility that investigators and educators shoulder to maximize the results of animal research through higher and more frequent reporting standards, challenging current trends in publication [32,33] and reproducibility [34]. For example, in spinal cord injury research alone, 2%–41% of the experiments utilizing animal models are unpublished, contributing to an overestimation of efficacy [33]. Preclinical rates of unreproducible research may exceed 50%. The losses are huge: an estimated USD 28 billion is spent in that country annually alone on non-reproducible research [34]. These challenges contribute not only to economic and knowledge loss but also to unnecessary animal use and the loss of animal life.

We suggest that researchers publish all the research data gathered from animal experimentation transparently and in a manner that is consistent with the FAIR share principles (Findable, Accessible, Interoperable, and Reusable) [35]. By embracing Reflection and the responsibilities that Reflection engenders, researchers can decrease unnecessary animal experimentation and transparently advance translational efforts. Progress toward this goal has already started; the National Institutes of Health (NIH) in the USA, for example, has made data sharing mandatory, underlying the important obligation researchers have when using animals in research.

Reflection and Responsiveness augment the original 3R framework, and can enhance the ethical and scientific rigor of animal use in science and education. These new additions are intended to be inspiring and straightforward to implement. By fostering the ongoing evaluation of research based on new evidence and ensuring that all data, regardless of outcome, are published transparently, these new principles can drive more humane and effective research practices.

## 5. Conclusions

Currently, Canada’s decentralized approach to animal research and teaching governance leaves research animals vulnerable to welfare infractions, especially in the private sector, and can cause confusion among researchers and educators. A federal Animals in Science Act for Canada is an ultimate goal that would close these gaps. In a stepwise approach to this important, however, burdensome goal in a time of fiscal austerity in this country, a greater emphasis on the three Rs for animal research—with the addition of Reflection and Responsiveness—could significantly elevate animal welfare in Canada and serve as a model for other countries whose own interests in protecting animals in research and education are in consideration.

## Figures and Tables

**Table 1 animals-14-02755-t001:** Animal research protection across provinces and territories. [AB: Alberta; BC: British Columbia; MB: Manitoba; NB: New Brunswick; NL: Newfoundland and Labrador; NS: Nova Scotia; ON: Ontario; PE: Prince Edward Island; QC: Quebec; SK: Saskatchewan; NT: Northwest Territories; NU: Nunavut; YT: Yukon].

	AB	BC	MB	NB	NL	NS	ON	PE	QC	SK	NT	NU	YT
Specific animal researchregulations under animalwelfare or animalprotection act													
Standalone act for animalsin research													
Reference CCAC guidelines													

## Data Availability

Not applicable.

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
