# Peer review of "Stepwise Imperatives for Improving the Protection of Animals in Research and Education in Canada"

_animals, 2024, doi:10.3390/ani14192755_

Round 1
Reviewer 1 Report
Comments and Suggestions for Authors
This is an interesting manuscript describing problems that are created in Canada due to the decentralized approach to regulating animal use in science. The authors clearly describe the legal inconsistencies and, making a step forward, they present their proposals. The creation of an Animal in Science Act is the first one followed by the introduction of two additional Rs to the traditional 3Rs framework - Reflection and Responsiveness. Concerning the additional 2Rs, I am very skeptical. There are already different voices proposing additional Rs, like Responsibility or even 12Rs ( C. B. Brink & D. I. Lewis, Animals 2023,13,1128). I am also wondering, Responsiveness is not achieved by following the ARRIVE guidelines? If yes, what is the need to add another R?
I would propose the authors better document the need for upgrading the 3Rs or even reconsider their proposal as far same results can be achieved by already existing tools (ARRIVE or PREPARE or OBSERVE guidelines).
Author Response
Thank you very much for taking the time to review this manuscript. Please find the detailed responses below and the corresponding revisions/corrections highlighted in the re-submitted files. |
|
Comments 1: This is an interesting manuscript describing problems that are created in Canada due to the decentralized approach to regulating animal use in science. The authors clearly describe the legal inconsistencies and, making a step forward, they present their proposals. The creation of an Animal in Science Act is the first one followed by the introduction of two additional Rs to the traditional 3Rs framework - Reflection and Responsiveness. Concerning the additional 2Rs, I am very skeptical. There are already different voices proposing additional Rs, like Responsibility or even 12Rs ( C. B. Brink & D. I. Lewis, Animals 2023,13,1128). I am also wondering, Responsiveness is not achieved by following the ARRIVE guidelines? If yes, what is the need to add another R? I would propose the authors better document the need for upgrading the 3Rs or even reconsider their proposal as far same results can be achieved by already existing tools (ARRIVE or PREPARE or OBSERVE guidelines). |
Response 1: Thank you for pointing this out. We agree that there are other voices proposing additional Rs and guidelines. However, the original 3R framework and the revised ARRIVE guidelines both focus on the design phase. The proposed new Rs go beyond this phase to target the entire animal use protocol, and the phase during which the research is conducted in particular. The additional Rs are meant to be easy to follow and promote continuous reflection throughout the experimental process and post-research. We have added additional clarification and referenced the ARRIVE guidelines specifically. Changes have been made directly in the manuscript in lines 209-217 and 241-244.
|
Reviewer 2 Report
Comments and Suggestions for Authors
Thank you for a very interesting and thought-provoking article, which challenges the current legislative provisions for protection of research animals in Canada. I think the proposal to add 2 extra 'R's to the well-known '3R's approach to research ethics is innovative, and encourages the researcher who is proposing to use animals to be more reflexive about that intended use, as well as promoting open science. As you have stated, this could be introduced as a precursor to legislative change . I have a few minor comments that I feel would enhance readability:
1. Lines 78-82 this sentence states ‘with direct implications for the ethical practice in Canada that may impact the numbers of animals used and reported, choices of animal models, and continuity of care for animals in research and in educational settings.’ I wonder if it could be rephrased as follows ‘with direct implications for ethical practice in Canadian research and educational settings that may impact the numbers of animals used and reported, choices of animal models, and continuity of care for animals.’?
2. Lines 119-120. I was unsure if this was the total number of animals used, or if it was referring to the number of animals involved in reported breaches of the CCAC code? I assume it is the former, but I think it could be worded more clearly.
3. Line 134. I think ‘Committees’ should be singular here?
4. Line 140. The statement ‘Efficacy and efficiency are diluted’ could be expanded to explain what this phrase means.
5. Tables 1 and 2 – the ticks are quite faint, and could be made clearer. Also, the regulations referred to should be listed with other references, rather than below the table (perhaps as a separate section – ‘Legislation’ – in the reference list instead of being dotted around amongst the other references?)
6. Line 192 – would it be better to name the future ‘Animals in Science’ Act with Animals in plural, rather than the ‘Animal in Science’ Act (singular)? (it is referred to in several places)
7. Line 215 refers to ‘Canada’s fractured landscape’ when I think this means ‘Canada’s fractured legislative landscape’ which would be more descriptive.
Author Response
1. Summary |
|
|
Thank you very much for taking the time to review this manuscript. Please find the detailed responses below and the corresponding revisions/corrections highlighted in the re-submitted files.
|
||
2. Point-by-point response to Comments and Suggestions |
||
Comments 1: Lines 78-82 this sentence states ‘with direct implications for the ethical practice in Canada that may impact the numbers of animals used and reported, choices of animal models, and continuity of care for animals in research and in educational settings.’ I wonder if it could be rephrased as follows ‘with direct implications for ethical practice in Canadian research and educational settings that may impact the numbers of animals used and reported, choices of animal models, and continuity of care for animals.’? |
||
Response 1: Thank you for suggesting how to make this sentence clearer. We have integrated your suggestion, and it can be found in the manuscript at lines 80-82.
|
||
Comments 2: Lines 119-120. I was unsure if this was the total number of animals used, or if it was referring to the number of animals involved in reported breaches of the CCAC code? I assume it is the former, but I think it could be worded more clearly. |
||
Response 2: Thank you for pointing this out. We have revised the wording to make it clear that it is all the animals reported to be used and not animals involved in breaches. The changes are addressed in lines 119-120.
|
||
Comments 3: Line 134. I think ‘Committees’ should be singular here? |
||
Response 3: Thank you. Change to “committee” made at line 134 of the manuscript.
|
||
Comments 4: Line 140. The statement ‘Efficacy and efficiency are diluted’ could be expanded to explain what this phrase means. |
||
Response 4: We have added to this sentence and expanded the idea at lines 140-141 in the manuscript.
|
||
Comments 5: Tables 1 and 2 – the ticks are quite faint, and could be made clearer. Also, the regulations referred to should be listed with other references, rather than below the table (perhaps as a separate section – ‘Legislation’ – in the reference list instead of being dotted around amongst the other references?) |
||
Response 5: Thank you for this feedback. We have changed the checkmarks to be more prominent in both tables. We have also added the references into the main bibliography.
|
||
Comments 6: Line 192 – would it be better to name the future ‘Animals in Science’ Act with Animals in plural, rather than the ‘Animal in Science’ Act (singular)? (it is referred to in several places) |
||
Response 6: Thank you. Change made at lines 185 and 249 in the manuscript.
|
||
Comments 7: Line 215 refers to ‘Canada’s fractured landscape’ when I think this means ‘Canada’s fractured legislative landscape’ which would be more descriptive. |
||
Response 7: This suggestion improves readability and clarity. Thank you. Change made in the manuscript at lines 208-209.
|
Reviewer 3 Report
Comments and Suggestions for Authors
Reviewer comments for manuscript ID animals-3201959 entitled ‘Stepwise Imperatives for Improving the Protection of Animals in Research and Education in Canada’
General comments
Use of animals in research is now a topic of debate worldwide due to the inconsistencies in the implementation of the 3R’s in the ethical framework. Most of the guidelines for use of animals in research are considered as guidance tools that are not laws and lack legislation. This has lead of inconsistencies in the implementation of animal welfare guidelines and ethics in animal research. A worldwide comprehensive guidelines need to be frame by international agencies like OIE and FAO that push for legislation as laws on use of animals in research uniformly in all member countries.
The manuscript underlines an important aspect of implementation of experimental animal care and ethics in research that needs a centralized approach and effective legislation for implementation as a law across Canada. The authors have comprehensively pointed out the situation of experimental care in research across all provinces in Canada to strengthen their viewpoint. The addition of two more ethical viewpoints to be included in the existing 3R’s to further strengthen them into 5R’s is rightly justified through effective research on the existing situation across Canada on the use of animals in research . The writing is flawless, concise and conveys the viewpoint clearly. I congratulate the authors for this brilliant write up and I hope this rekindles action on further work in the welfare of animals in research through an ethical framework across Canada. I recommend the publication of this manuscript.
Specific comments
Lines 26-27: Please rewrite as ‘The two new Rs -Reflection and Responsiveness where Reflection emphasizes more continuous and specific attention to….’
Author Response
1. Summary |
|
|
Thank you very much for taking the time to review this manuscript. We appreciate your feedback and comments. Please find the detailed response below and the corresponding correction highlighted in the re-submitted files. |
||
|
||
Comments 1: Lines 26-27: Please rewrite as ‘The two new Rs -Reflection and Responsiveness where Reflection emphasizes more continuous and specific attention to….’ |
||
Response 1: Thank you for this suggestion. We have, accordingly, changed the wording to address this comment at line 27 in the manuscript.
|